# Best Practices for the Management of Patients with Non-Tuberculous Mycobacterial Pulmonary Disease According to a German Nationwide Analysis of Expert Centers

**DOI:** 10.3390/healthcare11192610

**Published:** 2023-09-22

**Authors:** Gernot Rohde, Monika Eichinger, Sven Gläser, Marion Heiß-Neumann, Jan Kehrmann, Claus Neurohr, Marko Obradovic, Tim Kröger-Kilian, Tobias Loebel, Christian Taube

**Affiliations:** 1Pneumologie/Allergologie, Medizinische Klinik 1, Universitätsklinikum Frankfurt, Goethe-Universität, 60590 Frankfurt am Main, Germany; gernot.rohde@kgu.de; 2Translational Lung Research Center Heidelberg (TLRC), German Center for Lung Research (DZL), 69120 Heidelberg, Germany; monika.eichinger@med.uni-heidelberg.de; 3Department of Diagnostic and Interventional Radiology, Heidelberg University Hospital, 69120 Heidelberg, Germany; 4Department of Diagnostic and Interventional Radiology with Nuclear Medicine, Thoraxklinik, Heidelberg University Hospital, 69126 Heidelberg, Germany; 5Vivantes Klinikum Neukölln und Spandau, Klinik für Innere Medizin-Pneumologie und Infektiologie, 13585 Berlin, Germany; 6Department of Pneumology, Asklepios Lungenfachklinik München-Gauting, 82131 Gauting, Germany; 7Institute of Medical Microbiology, University Hospital Essen, University of Duisburg-Essen, 45147 Essen, Germany; jan.kehrmann@uk-essen.de; 8Robert-Bosch-Krankenhaus Klinik Schillerhöhe—Lungenzentrum Stuttgart, 70376 Stuttgart, Germany; claus.neurohr@rbk.de; 9Insmed Germany GmbH, 60549 Frankfurt am Main, Germany; marko.obradovic@insmed.com (M.O.); tim.kroegerkilian@insmed.com (T.K.-K.); tobias.loebel@insmed.com (T.L.); 10Department of Pulmonary Medicine, Ruhrlandklinik, University Hospital Essen, 45239 Essen, Germany

**Keywords:** NTM, interviews, Germany, treatment centers, pulmonary, management, best practices

## Abstract

Non-tuberculous mycobacterial pulmonary disease (NTM-PD) is a chronic inflammatory lung disease caused by infection with non-tuberculous mycobacteria (NTM). International guidelines provide evidence-based recommendations on appropriate diagnosis and treatment strategies, but there is a need for sharing day-to-day best practice between treatment centers to optimize patient care. This is particularly valuable for rare diseases like NTM-PD. In this cross-sectional analysis of NTM-PD management in Germany, medical and administrative staff from seven treatment centers were interviewed to identify best practice in the diagnosis, treatment, and ongoing management of patients with NTM-PD, including related hospital infrastructure and administration processes. A prioritization led to a collection of best practices for the management of patients with NTM-PD in Germany, which is presented here. Selected current best practices included performance of regular sputum tests for diagnosis, use of medical reports, and regular follow-up visits as well as increased interaction between physicians across different specialties. Future best practices that may be implemented to overcome current barriers comprised disease awareness activities, patient empowerment, and new approaches to enhance physician interaction. Challenges related to their implementation are also discussed and will help to raise disease awareness. The presented best practices may guide and optimize patient management in other centers.

## 1. Introduction

Non-tuberculous mycobacterial pulmonary disease (NTM-PD) is a chronic inflammatory pulmonary disease caused by infection with non-tuberculous mycobacteria (NTM). These mycobacteria are also known as environmental mycobacteria and are genetically related to *M. tuberculosis*. In contrast to *M. tuberculosis*, which is typically transmitted from person to person by aerosols, NTM are acquired via environmental sources [1,2,3]. As of 2020, approximately 190 species of NTM had been identified, but their pathogenic relevance for human disease is variable [4,5,6].

NTM-PD is a rare disease with an annual prevalence of 3.3–6.2 per 100,000 in EU5 countries (United Kingdom, Spain, Italy, France, Germany), but the general prevalence has been rising in recent years [7,8,9,10]. Comparable trends were also observed in Germany, where the estimated mean annual NTM-PD prevalence rate was 2.3 per 100,000 inhabitants in 2009 and increased to 3.3 in 2014 [8]. Predisposing factors for NTM-PD include structural lung diseases, such as chronic obstructive pulmonary disease or bronchiectasis. Further, genetic mutations, such as mutations in the *CF TRANSMEMBRANE CONDUCTANCE REGULATOR* (*CFTR*) gene, as well as immunosuppression and corticosteroids predispose patients to developing the disease [11,12].

In general, NTM-PD is associated with a high burden of disease, including a negative impact on health-related quality of life [13]. It can be progressive and even fatal, especially if severe forms of the disease are left untreated [11]. Reported five-year mortality rates vary between studied populations but can be as high as 42% for patients with *M. avium complex* infection [14]. Prompt diagnosis and rapid therapeutic response are important to secure optimum patient outcomes.

However, diagnosis of NTM-PD in clinical practice can prove challenging, as a number of NTM-PD symptoms overlap with the aforementioned comorbid conditions, with chronic cough, usually with sputum production, and shortness of breath being the most common. Additional symptoms include fatigue, malaise, weight loss, and night sweats. Nevertheless, not all of these symptoms are specific to NTM-PD; they can also occur in other chronic diseases [15]. The relative rarity of NTM-PD in comparison to other respiratory conditions means that the index of suspicion is usually low, resulting in delayed recognition and diagnosis. Consequently, patients may remain undiagnosed for some time or NTM-PD may be an incidental finding during investigation for other comorbid conditions [6,11,15,16,17,18]. Guidelines recommend a multiple-criteria approach to NTM-PD diagnosis: presence of clinical symptoms, radiological evidence of bronchiectasis with multiple small nodules or caverns, and microbiologic proof of NTM species, e.g., microbiologic NTM detection primarily through culture, are all required for robust diagnosis. While chest radiography or ideally high-resolution computed tomography (HRCT) should be applied for radiologic analysis, microbiologic diagnosis should be based on positive NTM culture from sputum samples or bronchial washes/lavages or biopsies [6].

The treatment of patients with NTM-PD is complex, and long-term treatment approaches are needed. Guideline-based therapy usually involves a multi-drug regimen of at least 12 months, while watchful waiting is only the preferred course of action in some instances. Consequently, NTM-PD is associated with a significant treatment burden [6,13,15]. Therefore, multidisciplinary teams are considered best placed to support patients, with adjunctive therapies alongside antibiotic regimens, as necessary [19]. However, studies have shown that overall adherence to guideline recommendations during management of NTM-PD is low, including in Germany [20]. Further, a substantial proportion of German patients with NTM-PD discontinue therapy prematurely [21]. It was hypothesized that there are also infrastructural and management hurdles that influence best possible disease management. This analysis was performed in German expert centers to reveal putative barriers and identify best practices. Implementing the identified best practices—even outside of specialized NTM-PD expert centers—and raising awareness for existing barriers may help to overcome current widespread limitations in NTM-PD management and consequently improve patient care.

## 2. Materials and Methods

### 2.1. Interviews

To identify best practices as well as challenges and barriers in the management of patients with NTM-PD, seven German expert centers were evaluated. Expert centers represent experienced NTM treatment hospitals based on their number of managed NTM-PD patients per year. To minimize putative regional differences and receive feedback from all parts of the country, the selected expert centers were spread all over Germany. Structured interviews via phone or videocall were performed from 2018 to 2022 in the following centers: University Hospital Essen—Ruhrlandklinik, Universitätsklinikum Frankfurt, Thoraxklinik Heidelberg, Vivantes Klinik Berlin, Lungenfachklinik Immenhausen, Robert-Bosch-Krankenhaus Lungenzentrum Stuttgart, and Asklepios Lungenfachklinik Gauting. Interviewees were staff members from different specialties and departments, including pulmonology, radiology, microbiology (laboratories), and management/administration. Appropriate interviewees were identified within the centers and referred to the data collection agency by the chief physician. Where possible, at least one representative per center was interviewed, with an average of five interviewees per center. Interview guides applied for different specialties are provided in the Appendix A. Insmed Germany GmbH was involved in the generation of the interview guides. Interviews and analysis of interview data were conducted by a data collection agency, independent of the sponsor.

Paper-based interview transcripts were electronically filed in one center for further data processing after all interviews had been completed. The data collection agency evaluated the interviewees’ statements in a first step for each center. To this end, observations were labeled either as well-running processes (best practice) or challenges/barriers. During a debrief in cooperation with the centers, this categorization was reviewed and putative solutions to overcome barriers were discussed. Interviews and analyses of interviewees’ statements were carried out independently by two different individuals.

### 2.2. Transversal Analysis

A transversal analysis was carried out to quantitatively assess the interview results and to identify common best practices and challenges/barriers across all expert centers. First, the interviewees’ statements were summarized and phrased as practice elements. All elements that were mentioned by ≥1 center during the interviews were included in the transversal analysis. Elements that were considered to be beyond the influence of the centers (e.g., because of legal requirements) or referred to routine practices (i.e., standard procedures according to center-specific rules or treatment guidelines) were excluded from further analysis. Second, elements were categorized either as “best practices” (if scored as an optimally running working process), as “possible best practice” (if mentioned as a future solution to overcome a current barrier), or as a challenge/barrier. For clarification, possible best practices had not been established in any of the centers at the time of the interview. Next, practice elements (best practices, possible best practices, and challenges/barriers) were ranked per category according to their mentions across all centers to quantitatively evaluate the interview results.

### 2.3. Survey and Workshop for Expert Evaluation of Transversal Analysis Results

For further prioritization and semiquantitative analysis of the transversal analysis results, an expert panel was assembled and surveyed using the Mentimeter interactive survey tool (https://www.mentimeter.com/, accessed on 23 June 2022). Mentimeter is an interactive presentation software package that enabled real-time and anonymous voting and direct interaction among the expert panel. All voting was single-blinded, i.e., experts were blinded against the others’ votes. The anonymized voting results were subsequently discussed. Experts have approved use of the mentimeter tool upfront. The expert panel consisted of 7 experts, with one representative each from the NTM-PD expert centers in Frankfurt, Heidelberg, Berlin, München, and Stuttgart and two representatives from Essen. To ensure multidisciplinarity, experts from all specialties and departments that had been interviewed (excluding management/administration) were selected to participate in the expert panel. Best practices and possible best practices that achieved a ≥80% consensus among the expert panel in the online survey were selected for best practice sharing (based on a Delphi Consensus model) [22,23]. Those elements that received a 50 to 79% consensus in the online survey were selected for open discussion followed by re-evaluation by the same panel in an online workshop; this step was included to ensure that all elements reaching majority consensus were carefully considered, even if they did not achieve 80% consensus. Elements with less than 50% consensus were excluded from further evaluation. Best practices and possible best practices that achieved a majority consensus after re-evaluation during the workshop were finally also selected for best practice sharing. Subsequently, these selected elements were named “current best practices” (previous designation “best practices”) and “future best practices” (previous designation “possible best practices”).

In addition, current and future best practices were also evaluated for their relative feasibility and impact using a voting exercise with the Mentimeter live polling tool (https://www.mentimeter.com/, accessed on 23 June 2022). Specifically, experts organized practices according to their feasibility (high or low) and their impact (high or low) in a 2 × 2 matrix. Aspects that were considered for feasibility and impact evaluation comprised financial and human resources to implement/carry out these best practices and their putative positive impact on patient management. Current and future best practices are presented in a table and summarized in a figure.

Challenges and barriers, as identified in the transversal analysis, were re-evaluated by the expert panel during the workshop and were ranked according to their relevance in clinical practice using the Mentimeter live polling tool (https://www.mentimeter.com/, accessed on 23 June 2022). This voting stimulated an open discussion, where experts compared the relevance of each challenge/barrier in the context of the others and shared center-specific experiences. Selected challenges and barriers are presented in Table 1.

A schematic overview of the methodology is provided in Figure 1.

### 2.4. Data Reporting

The results from the online survey and discussions during the workshops were consolidated by the authors and shared among the expert panel for further feedback. As illustrated in Figure 1, the consolidated feedback led to the selection of current and future best practices, as well as challenges/barriers, which are presented in this publication categorized by areas of interest.

## 3. Results

### 3.1. NTM-PD Expert Centers Overview

Seven German NTM-PD expert centers were analyzed via structured telephone or videocall interviews from 2018 to 2022. These centers have broad expertise in the treatment of lung diseases, specifically in the management of NTM-PD. All centers have multidisciplinary teams in place for NTM-PD management. They are spread all over Germany to reflect putative differences in the disease management according to their geographical location. The mean number of patients treated annually per center was 48 (a range of 15–100 patients/year), of which 30% of cases were primary (i.e., new) NTM-PD diagnoses. About two thirds were patients with chronic NTM-PD. Overall, 50% of patients received treatment, while the rest either refused treatment or were managed by watchful waiting. All centers offered inpatient and outpatient services, including specialized outpatient care [26]. Primary patient contact usually occurred via outpatient clinics, and patients were referred for further inpatient treatment if indicated.

### 3.2. Transversal Analysis Results

Transversal analysis of interview data led to the identification of 34 practice elements, which could be assigned into seven general areas of interest, as illustrated in Figure 2. These areas were defined to allow clustering of elements and reduce complexity. Four areas of interest referred to the clinical phases in NTM-PD management (a), referral, diagnosis, therapy, and follow-up, and three areas of interest could be categorized as aspects of infrastructure for NTM-PD management (b), organizational structures, finances, and communication.

In total, 14 elements were categorized as best practices and 10 elements each were classified as possible best practices or challenges/barriers. While the importance of early referral and close follow-up were highlighted in the selected (possible) best practices, underfunding and lack of staff resources were the major obstacles for improved NTM_PD management. All practice elements (N = 34) as well as their purposes are shown in Table 1 and Table 2.

The complete list of the 34 identified elements with a detailed description of each element can also be found in Appendix A.

### 3.3. Selection of Current and Future Best Practices

Further prioritization of practice elements by the assembled expert panel led to the selection of 10 current and 9 future best practices, which are summarized in Figure 3. The list of selected and rejected best and possible best practice elements can also be found in Appendix A.

#### 3.3.1. Clinical Phases in the Management of NTM-PD (a)

##### Referral

For the referral phase of the NTM-PD management process, one current best practice was identified:Direct patient transfer to specialized centers.

Future best practices in this phase included:
II.An information flyer for the referring and/or treating physicians.III.NTM-PD awareness activities combined with information activities for tuberculosis.

In line with its classification as current best practice, practice element I in this category was classified as having the highest impact and feasibility, while future best practices (II and III) were evaluated as having average impact and feasibility, with the lowest feasibility for NTM-PD awareness activities.

##### Diagnosis

For the NTM-PD diagnosis, three practice elements were identified as current best practices:I.Performance of a regular sputum test (i.e., once per year in high-risk populations or in case of typical NTM-PD symptoms) [24,25].II.Use of HRCT imaging [6].III.Use of established communication channels for queries (by phone call or email).

Further, one future best practice was identified:IV.Establishment of protocols (standard operating procedure, etc.).

Current best practices I and II were evaluated as having the highest impact and feasibility. This also corresponds to their classification as guideline and consensus recommendations. Performance of a sputum test may be implemented as part of the regular follow-up visit. HRCT imaging is available in all centers and provides the possibility to assess radiographic disease progression. In contrast, the use of established communication channels for queries to enable correct and guideline-based NTM-PD diagnosis was considered to have high impact, but its feasibility is hindered by limited time resources. Establishing standardized protocols (IV) that would standardize management approaches for all physicians in the center and also provide guidance for new and less experienced healthcare professionals who join the team was categorized as feasible, and may have a positive impact on NTM-PD diagnosis in future.

##### Therapy

For NTM-PD therapy, one current best practice was identified:I.Use of medical reports (*Arztbriefe* in German)

Moreover, there was also one future best practice in this clinical phase that was selected for best practice sharing:II.Focus on patient empowerment, either through homework, training, or knowledge sharing (such as publications).

According to its classification, the current best practice was of significantly higher feasibility than the identified future best practice. The impact of both practices might be comparable. Medical reports (*Arztbriefe* in German) are the main source of information about a patient’s status, including medical history and diagnoses. As the German healthcare system is not fully digitized yet, these reports are crucial for best possible patient management, especially if different physicians are involved. Practice element #10 from the shortlist of best practices after the transversal analysis was rejected, as a hotline for better patient support would be challenging, highly time-consuming, and too expensive to implement given the lack of reimbursement. In general, it is difficult to justify the cost of a service, such as a patient hotline, given the small patient population with NTM-PD.

##### Follow-Up

Finally, two practice elements were selected for the follow-up phase; while element I was identified as current, element II was categorized as a future best practice:I.Follow-up of patients for regular check-ups every 3 to 6 months.II.A cross-center bronchiectasis and/or NTM-PD registry for monitoring of the patient population.

The feasibility and impact of the current best practice (I) was evaluated as very high, while the future best practice (II) was classified as having much lower feasibility. As was revealed during open discussion and based on previous experiences, low feasibility of element II can mainly be attributed to challenges in the data collection process, especially if different hospitals with variable documentation systems are involved.

#### 3.3.2. Infrastructure in the NTM-PD Management Process (b)

##### Organizational Structures

In total, two current best practices associated with organizational structures in NTM-PD management were identified:I.Internal lung boards (specialist conferences).II.An affiliation of specialized outpatient care [24].

In addition, one future best practice was selected for this category: III.Implementation of nationwide lung boards (training events).

Current best practice I was classified as having the highest impact and feasibility, followed by current best practice II. During open discussion, it was revealed that tumor review boards in oncology may serve as a blueprint for specialist conferences, since these structures have been found to be invaluable in optimizing patient care. The implementation of nationwide lung boards (III) grouping lung specialists across Germany in a virtual event in future might be challenging, but was considered highly valuable to discuss difficult patient cases and share knowledge and learnings. Best practice elements #16 and #17 from the transversal analysis shortlist were rejected due to different reasons. Best practice element #16 was considered to be redundant due to current best practice II in this category and, moreover, its implementation is not feasible due to German regulations about healthcare providers. Best practice element #17 was, on the one hand, referred to as a routine practice since internal electronic recording of patient history is a standard across German hospitals. On the other hand, implementation of the element is currently impossible as an electronic information system is not available in all hospitals.

##### Communication

Current best practices related to communication in the NTM-PD management process were:I.Tele-radiology as part of regular multidisciplinary NTM-PD boards, which should include experts from different disciplines, such as respiratory specialists, infectious disease specialists, microbiologists, radiologists, and, if needed, pharmacists and thoracic surgeons for complex cases.II.Exchange of images by radiologists directly via an online interface.

To optimize communication in future, the following possible best practices were selected:III.A virtually shared network for referring physicians and clinicians.IV.A specialized team with clear responsibilities for patient care.

The highest feasibility and impact scores were attributed to current best practices I and II, while future best practices II and IV were considered as having significant impact but low feasibility. Best practice element #22 identified in the transversal analysis was rejected as this communication tool is currently only rarely used.

##### Finances

No current best practice for the financial management of NTM-PD across the evaluated treatment centers was identified as practice element #24 identified in the transversal analysis was rejected due to low feasibility. One element was selected as a future best practice: better explanation and filing of previous findings.

### 3.4. Challenges/Barriers in NTM-PD Management

A lack of resources, including staff, financial investment, and time, was identified as a general barrier that is influential at all stages of clinical and structural NTM-PD management and is therefore related to all challenges identified in the different categories.

#### 3.4.1. Clinical Phases in NTM-PD Management (a)

During patient referral, incomplete documentation of findings and result reports was considered the most relevant barrier in this category. Further, limited experience of the referring physician and no specific request for clarification of NTM infection during referral were current barriers that were identified across all expert centers but were of lower relevance. Restriction of patient referral to pulmonary specialists only was categorized as a challenge specific to the German healthcare system and was consequently evaluated as of lowest relevance.

Limited resources were identified as key challenges in the diagnostic process. Since NTM-PD is a highly variable, multifactorial, and complex disease, treatment decisions need to be taken at an individual level. Consequently, this represents a highly time-consuming process that involves staff from various disciplines. Furthermore, based on this complexity of disease management, regular feedback with the referring physician was identified as an additional challenge in the context of patient discharge and follow-up. The lack of adherence to guideline-based therapies and possibly also to specified therapy plans was another complicating factor in this context, which was evaluated as having the highest relevance in these categories.

#### 3.4.2. Infrastructure for NTM-PD Management (b)

Effective external communication in the management of NTM-PD is currently hampered by the lack of a suitable communication channel for the exchange of patient information, results, and diagnoses. However, this barrier can be overcome and therefore was evaluated as being of lower general relevance. Financial issues that negatively affect NTM-PD treatment in Germany were complex and error-prone processes to invoice inpatient and outpatient services and almost no in-house expertise available for billing according to the uniform evaluation standard (*Einheitlicher Bewertungsmaßstab* in German). The former challenge was evaluated as being of highest relevance in the NTM-PD center infrastructure, while the latter was negligible.

This structured analysis of the management of NTM-PD patients across German reference centers led to the identification of 10 current best practices. These reflect optimally running working processes, including a systematic approach for NTM-PD diagnosis (regular sputum testing, use of HRCT imaging) and measures to enhance physician interaction (during referral, diagnosis, therapy, and follow-up). In addition, nine future best practices were identified whose implementation is currently limited by existing obstacles. These future best practices referred mostly to a stronger interaction between physicians. Finally, 10 challenges and hurdles were revealed in the context of this analysis.

## 4. Discussion

### 4.1. Best Practices in the NTM-PD Management Process and Challenges Associated with Their Implementation

In this study, 10 current and 9 future best practices, as well as 10 challenges/barriers, were identified across the evaluated German NTM-PD expert centers.

#### 4.1.1. Best Practices during Patient Referral

Direct transfer of patients to specialized centers was selected as a current best practice in the referral phase of patients with NTM-PD. However, NTM-PD expert centers may include different organizational structures, such as centers linked to university hospitals or specialized outpatient care [24]. The different outpatient settings, defined according to German regulations, vary in their complexity and bureaucracy and offer different advantages and disadvantages. However, providing high-quality patient care can be achieved within various organizational settings. Therefore, the selection of the appropriate (outpatient) care setting should be performed according to the hospital’s individual needs and local factors.

#### 4.1.2. Best Practice Diagnostics

Performance of regular sputum tests for patients at risk of NTM infection was identified as another current best practice in NTM-PD diagnosis. In order to establish this measure as a routine procedure in all treatment centers, the system of regular sputum testing for follow-up of NTM-PD patients under treatment or watch and wait supervision may serve as a blueprint: patients may receive sample tubes at their regular visits and samples may be directly sent to specialized laboratories at NTM-PD expert centers. This should enable existing infrastructure to be used most efficiently and the number of sputum tests should therefore increase.

Future best practices identified in the diagnosis phase should help to increase disease awareness and expertise, leading to an earlier referral and NTM-PD diagnosis [27]. In this analysis, limited experience from the referring physician was identified as a key barrier in early NTM-PD management. Further, late diagnoses due to unspecific symptoms or due to an overlap in symptoms with underlying lung disease are common problems for NTM-PD in general [28]. The need for heightened awareness and increased knowledge of NTM-PD was also one of the key outcomes of the ENPADE (European NTM-PD patient disease experience) survey [29]. However, implementation of the identified future best practices is likely to be associated with a considerable effort, as is analyzing the impact of these implemented practices. For instance, it would be difficult to assess the impact of information brochures (digital or print) on patient referral to specialized centers. NTM-PD awareness activities might include training for physicians in private practices or improved medical education through symposia and other educational events. Patient education and raised awareness around the prevalence and symptoms of NTM-PD could also be valuable, particularly in high-risk populations. Disease awareness campaigns have been shown to significantly increase public knowledge of disease symptoms and warning signs [30], which, in the case of NTM-PD, could reduce the delay between initial development of symptoms and receipt of appropriate treatment. However, the lack of a dedicated and defined concept for how such training and awareness campaigns should be coordinated between the individual expert centers resulted in lowest feasibility values in the respective analysis. This implies that these respective future best practices, although challenging, will bring benefits by tackling certain current problems.

To overcome obstacles in the use of established communication channels in NTM-PD diagnosis, new and innovative measures might be taken in future. These may include a telephone hotline or an online platform, the use of messenger apps for physician communication, or specialized NTM-PD outpatient clinics or nurses. Specialized nurses in interstitial lung disease centers have already been shown to improve overall patient care [31]. In this analysis, a dedicated team with clear responsibilities was identified as a future best practice to optimize the communication, but feasibility values were low due to limited staff resources. Enhanced cooperation across the different disciplines within a center, e.g., when convening lung and NTM-PD boards, was a key component of current best practices in the center infrastructure. These practices should also be complemented by the use of appropriate technologies, such as online interfaces or via virtual networks in future. An electronic documentation system for all patient-specific data is the prerequisite for effective NTM-PD patient management and has already been established in all centers.

In general, all investments in and financial efforts towards current and future measures need to be weighed against the potential benefits, particularly since NTM-PD is a rare disease [7,8]. Systemic underfunding and lack of resources are major obstacles for improved NTM-PD management. With no financial incentives from the healthcare system, the main burden to applying optimum NTM-PD care falls to personal efforts by committed individuals within centers. Given these financial restrictions, patients may be referred to alternative specialized centers with a longer history and well-established structures, e.g., cystic fibrosis centers, as an alternative to specialized NTM-PD outpatient clinics. Further, these may also serve as a blueprint with regard to NTM-PD center infrastructure and organization.

#### 4.1.3. Best Practices during Therapy

The use of medical reports was considered best practice during treatment provision. These reports are invaluable in sharing clinical history and outcomes both within and outside the patient’ treatment team, particularly as not all healthcare systems currently benefit from fully digitized systems (which could achieve similar results). For the management of more complex patient cases, where the specialist support is required that is beyond the expertise of the microbiologist, clinicians in Germany can also receive additional support from the National Reference Center for Mycobacteria [32]. This center offers a variety of services, including microbiological evaluation of sputum and bronchoalveolar lavage samples, second opinions, and antibiotic susceptibility testing. Further queries with regard to the prevention, diagnosis, and therapy of mycobacterial infections can also be addressed to experts of the German Center for Infection Research (DZIF), who offer a telephone hotline [33]. Precedence for the success of such knowledge-sharing networks includes the tumor review boards used across oncology [34], which can provide valuable support to clinicians during diagnosis and management of complex patient cases. However, the field of oncology benefits from greater access to the resources necessary to develop and implement this type of approach, due to the relatively large patient populations. Replicating this model for NTM-PD could prove more challenging, although perhaps pilot testing at a smaller scale could be considered in future. To identify NTM-PD expert centers that can support the diagnosis and management of patients with NTM-PD, referring physicians may also make use of online platforms, such as Orphanet and SE-ATLAS (*Versorgungsatlas für seltene Erkrankungen* in German) [35,36].

Another consideration in NTM-PD treatment decisions is the relative benefits of early treatment initiation vs. watch and wait supervision. Some evidence suggests that NTM-PD can result in clinical deterioration and worsening patient outcomes if left untreated [11]. Guidelines suggest that prompt treatment may moderate such disease progression and improve patient outcomes [6]. However, individual patient factors such as age, virulence of NTM species, and stability of symptoms should also be considered given the burden of treatment.

The role of the patient could also be acknowledged and fortified through the use of patient support programs, including medication management and counselling offered by nurses or phone services or provided by online resources, which have been shown to have a positive impact on therapy adherence in chronic diseases [37,38]. During the COVID-19 pandemic, an app-based care program for German patients with chronic and acute lung disease had positive impacts on the patient’s quality of life and increased therapy adherence [39]. A potentially valuable objective, a center-specific telephone hotline for better patient support in the context of the NTM-PD therapy was rejected from the list of best practices during this study due to a lack of financial resources.

Overall, limited resources were identified as key barriers in all phases of NTM-PD management, and the center infrastructure and limited funding were the most critical ones in the outpatient setting. Consequently, they also represent the main obstacles hampering the establishment of a cross-center bronchiectasis and/or NTM registry, which was identified as a future best practice in the follow-up phase. Although there is an overlap in the bronchiectasis and NTM-PD patient populations, two separate registries for both entities might be established, as therapy courses are different [26]. However, while a nationwide or international NTM-PD registry has not been implemented to date, a center-specific registry is currently under construction at the university hospital in Frankfurt. The Rhein-Main NTM Register is an investigator-initiated research project, which will provide a basis for evaluation and optimization of diagnosis and treatment of NTM-PD in the Rhein–Main area of Germany. A central database allows for the development of a prospective, longitudinal register of patients with NTM-PD and drives systematic collection of clinical data. Such data include demographics, comorbidities and risk factors, medication, results of physical examination, blood laboratory, lung function, radiology, detailed NTM treatment course, and health-related quality of life. A biobank for blood, urine, respiratory specimens, and NTM isolates is also planned for construction. The Register’s primary objective is to identify independent predictors of outcomes in patients with diagnosed NTM infection and additionally to study the epidemiology of NTM infections and obtain real-life data regarding the current management in the Rhein–Main area. The combination of the biobank with well-matched clinical data will additionally foster both basic and translational NTM research. Experiences from this registry, including the type and format of deposited patient data, might be transferred to a nationwide registry in future. Funding would be required to support such a registry.

#### 4.1.4. Best Practices during Follow-Up

All patients with NTM-PD should be followed up with at regular time intervals, and management of adverse events and testing for antibiotic resistance should be standard procedures. Monthly collected sputum samples can be used to document the treatment success and culture conversion. In order to counteract the impact of NTM-PD on social and emotional wellbeing, as also identified in the ENPADE survey, conservative therapy approaches to mobilize respiratory secretions and the patient’s need for psychological support, rehabilitation, and physiotherapy depending on disease characteristics should also be assessed at these regular patient visits [29].

#### 4.1.5. Evaluating the Implementation of Best Practices

Implementation of these recommendations across the German healthcare setting could serve to align clinical practice, with the ultimate aim of improving patient outcomes. However, assessment of the impact of such interventions could prove challenging. A randomized controlled trial would be the most robust evaluation method, but a substantial patient cohort would be required to achieve sufficient statistical power. A registry could support assessment of whether future best practices will have a qualitative impact on outcomes such as treatment adherence and persistence.

### 4.2. Study Limitations

This study represents a Germany-wide analysis of selected NTM-PD expert centers, with the implication that the identified best practices, challenges, and barriers were partially related to country-specific regulations and circumstances. Selection criteria for the NTM-PD expert centers were rather unspecific and included an interdisciplinary team structure of at least respiratory specialists, microbiologists, and radiologists, an outpatient setting, and >15 patients with NTM-PD/year. However, as shown by the interviews and transversal analysis, there was a considerable overlap between the best practices in the individual centers. Interviews and data analysis were performed from 2018 to 2022 and impacted by restrictions due to the COVID-19 pandemic. This fact raises the possibility that centers may have since implemented changes that were not captured during the data collection process.

### 4.3. Key Findings and Implications for Practice

Prompt transfer of patients with NTM-PD to specialized centers is recommended during the early stages of disease management. Regular sputum tests in accordance with guidelines should be performed in all patients at risk of NTM infection and/or demonstrating signs or symptoms of NTM-PD.

The index of suspicion of NTM-PD is low due to the rarity of the disease and lack of physician awareness of its prevalence and potential clinical presentation. Physicians should monitor patients with comorbid conditions for signs of NTM infection and initiate the guideline-based diagnostic process as applicable. Increased awareness of NTM-PD could be achieved through medical education and public awareness campaigns.

More effective communication between physicians, including the continued use of medical reports, would optimize patient outcomes. Direct support of patients through support programs and clear communication channels could also be beneficial.

Registries would allow for long-term outcome monitoring and prediction. The Rhein-Main Register in Frankfurt may serve as a blueprint for future endeavors. Patients should be followed up with at regular intervals, with monitoring of both clinical outcomes and mental wellbeing recommended, given the burden of NTM-PD treatment.

Most current and future best practices are limited by a lack of available resources at all levels of the healthcare system.

## 5. Conclusions

This study provides guidance on best practices that could be applied by hospitals and community healthcare practitioners in managing patients with NTM-PD. Best practices were identified through structured analysis of German NTM-PD expert centers, including structured interviews followed by an online survey and workshop for further evaluation by practicing healthcare professionals. Specialized expert centers are central players and crucial for best care of patients with NTM-PD. Consequently, they are increasingly being established. However, there are organizational and financial barriers that need to be overcome to increase disease awareness and to enable implementation of the presented best practices in future. In addition, patients and referring physicians need support to keep an overview of already-existing expert centers for rare lung diseases, such as NTM-PD.

## Figures and Tables

**Figure 1 healthcare-11-02610-f001:**
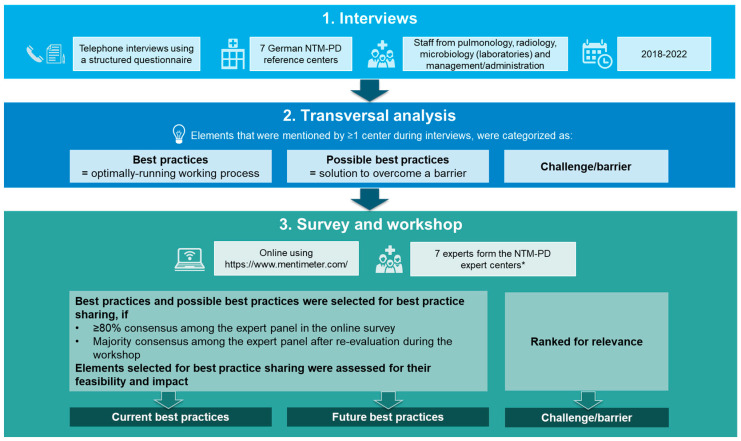
Overview of study methodology. * One representative each from the NTM-PD expert centers in Frankfurt, Heidelberg, Berlin, München, and Stuttgart, and two representatives from Essen. NTM-PD, non-tuberculous mycobacteria pulmonary disease.

**Figure 2 healthcare-11-02610-f002:**
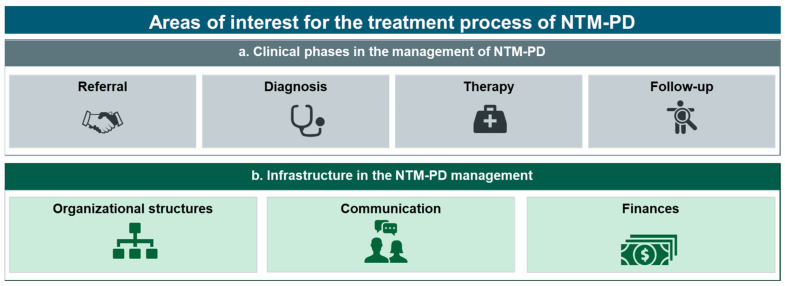
Elements identified in the clinical phases and infrastructure for NTM-PD management. NTM-PD, non-tuberculous mycobacteria pulmonary disease.

**Figure 3 healthcare-11-02610-f003:**
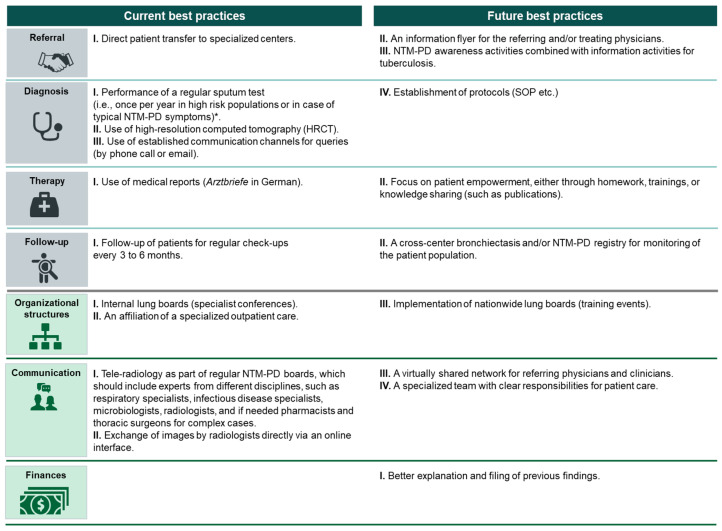
Current and future best practices selected for best practice sharing. * For more information, please refer to guidelines [6]. HRCT, high-resolution computed tomography; NTM-PD, non-tuberculous mycobacteria pulmonary disease; SOP, standard operating procedure.

**Table 1 healthcare-11-02610-t001:** Best practices (BPs) and possible best practices (PBPs) (N = 24) identified during transversal analysis, which were presented to the expert panel for discussion and selection.

Element No.	a. Clinical Phases in the Management of NTM-PD	Category
	**Referral**	
1	I. Direct patient transfer to specialized centers after patient referral.	BP
2	II. An information flyer for the referring and/or treating physicians to gain more knowledge on NTM-PD.	PBP
3	III. NTM-PD awareness activities could be combined with information activities for tuberculosis.	PBP
	**Diagnosis**	
4	I. Performance of regular sputum test (i.e., once per year in high-risk populations or in cases of typical NTM-PD symptoms, such as increase in coughing and amount of sputum, weight loss, and fever/night sweats; bronchoalveolar lavage (BAL) should be tested for acid-fast bacilli (AFB) in suspected cases) * [24,25].	BP
5	II. Use of high-resolution computed tomography (HRCT) imaging for higher precision in the diagnosis [6].	BP
6	III. Use of established communication channels for queries (by phone call or email) during the diagnostic process to enable correct and guideline-based diagnosis of NTM-PD.	BP
7	IV. Establishment of protocols (standard operating procedures, etc.) to support the diagnostic process.	PBP
	**Therapy**	
8	I. Use of medical reports (*Arztbriefe* in German) to share recommendations of action with the referrer	BP
9	II. Focus on patient empowerment, either through homework, training, or knowledge sharing (such as publications), to raise treatment engagement.	PBP
10	III. Individual hotline for better support of patients. Patients will be able to make telephone calls at any point in time with their questions concerning the therapy.	BP
	**Follow-up**	
11	I. Follow-up of patients for regular check-ups every 3 to 6 months *.	BP
12	II. A cross-center bronchiectasis and/or NTM-PD registry for monitoring of the patient population.	PBP
	**b. Infrastructure in NTM-PD management**	
	**Organization structures**	
13	I. Internal lung boards (specialist conferences) for better communication and training, e.g., in the diagnostic process to discuss unclear cases.	BP
14	II. An affiliation of specialized outpatient care.	BP
15	III. Implementation of nationwide lung boards (training events) to share awareness and engage referrers and especially general practitioners (GPs) due to their key role as “entry gatekeeper”.	PBP
16	IV. Merging of tuberculosis and NTM-PD specialized outpatient care.	BP
17	V. Internal electronic recording of patient history (*Electronic Recording System*) to go directly into the hospital information system.	BP
	**Communication**	
18	I. Tele-radiology as part of regular NTM-PD boards (composed of experts from different disciplines, such as respiratory specialists, infectious disease specialists, microbiologists, radiologists, and, if needed, pharmacists and thoracic surgeons for complex cases).	BP
19	II. Exchange of images by radiologists directly via an online interface for effort and time savings.	BP
20	III. A virtually shared network for referring physicians and clinicians to exchange patient information would also save effort and time in cases where second opinions are required.	PBP
21	IV. A specialized team with clear responsibilities for patient care to avoid confusion of external communications around the patient history, ease the effort and time spent on internal alignment, and offer better support to the patient.	PBP
22	V. Use of an app as a means of communication between medical specialists.	PBP
	**Finances**	
23	I. Better explanation and filing of previous findings to possibly save some costly investigations	PBP
24	II. Documentation of all services rendered and forwarding of these to the medical administration department.	BP

* For more information, please refer to guidelines [6]. AFB, acid fast bacilli; BAL, bronchoalveolar lavage; BP, best practice; HRCT, high-resolution computed tomography; GP, general practitioner; NTM-PD, non-tuberculous mycobacteria pulmonary disease; PBP, possible best practice.

**Table 2 healthcare-11-02610-t002:** Challenges/barriers identified during transversal analysis (N = 10), which were presented to the expert panel for discussion and selection.

Element No.	a. Clinical Phases in the Management of NTM-PD
	**Referral**
1	Limited experience of referring physician (including general physicians and respiratory specialists) with diagnosis and management of NTM-PD.
2	No specific request for investigation of NTM-PD infection during referral, leading to most patients being incidental findings.
3	Incomplete documentation of findings and result reports, leading to increased effort and complications.
4	Referral is only possible through a respiratory specialist, which can lead to extended waiting times.
	**Diagnosis**
5	Lack of time and staff to make diagnostic and treatment decisions, as the disease is individual and complex.
	**Follow-up**
6	Time-consuming regular feedback with the referring physician.
7	Lack of adherence to guideline-based therapies and possibly also to specified therapy plans.
	**b. Infrastructure in NTM-PD management**
	**Communication**
8	Lack of a suitable external communication channel for the exchange of patient data, findings, and diagnoses.
	**Finances**
9	Complex and time-consuming processes for services billed both in the inpatient and outpatient setting, which can lead to mistakes.
10	Almost no in-house expertise available for billing via a uniform evaluation standard (*Einheitlicher Bewertungsmaßstab* in German) despite a very complicated system.

NTM-PD, non-tuberculous mycobacteria pulmonary disease.

## Data Availability

Raw data supporting the analyses presented in this manuscript cannot be shared.

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
