# Peer review of "Best Practices for the Management of Patients with Non-Tuberculous Mycobacterial Pulmonary Disease According to a German Nationwide Analysis of Expert Centers"

_healthcare, 2023, doi:10.3390/healthcare11192610_

Round 1

Reviewer 1 Report

This research clarifies the prioritized selection of best practices for the treatment of patients with non-tuberculous mycobacterial pulmonary disease (NTM-PD) in Germany. NTM-PD is a rare disease, thus this research has guiding significance for managing patients with NTM-PD in clinical practice. Overall, this manuscript is generally written clearly and concise. However, there are some aminor problems  that should be clarified to enable publication.

 Minor points

  1. For "clinical phases in the management of NTM-PD (a)" in the Result Section,   regular interval s of  3 to 6 months was recommeded in the follow-up phase .  However, the content of follow-up algorithm, such as  appropriate  intervals of  sputum tests and CT scaning , should be adrressed. 
  2.  The conclusions need to  be presented in a clear and concise style in the Discussion. 

Author Response

This research clarifies the prioritized selection of best practices for the treatment of patients with non-tuberculous mycobacterial pulmonary disease (NTM-PD) in Germany. NTM-PD is a rare disease; thus, this research has guiding significance for managing patients with NTM-PD in clinical practice. Overall, this manuscript is generally written clearly and concise. However, there are some minor problems that should be clarified to enable publication.

Many thanks for this positive feedback. We very much appreciate the time you have taken to review our manuscript.

Results

1.       For "clinical phases in the management of NTM-PD (a)" in the Result Section, regular intervals of 3 to 6 months were recommended in the follow-up phase. However, the content of follow-up algorithm, such as appropriate intervals of sputum tests and CT scanning, should be addressed.

To highlight that treatment recommendations from guidelines should be followed, we added the following footnote in Table 1: “*For more information, please refer to guidelines.”

Discussion/Conclusions

2.      The conclusions need to be presented in a clear and concise style in the Discussion.

Thank you for this valuable comment. We have updated the discussion section to state and expand upon our conclusions more clearly.

Reviewer 2 Report

This paper is about best practice in non-tuberculous mycobacterial pulmonary disease. It is very concise in methodology, summary, etc. It deserves to be published after some revisions.

After some revisions, it deserves to be published.

Abstract

Please provide a brief summary of best practice in the abstract.

Introduction

Please add specific country names for the EU5 countries as a supplement.

Materials and Methods

No modification is required.

Results

In Figure 3, as in Table 1, is it recommended to perform the regular sputum test once a year in the high-risk population?

Discussion

In this study, the timing of treatment initiation and treatment regimen (antimicrobial regimen) are not included in Best Practices. Was this discussed? Why was this not included in Best practices? Please state in the discussion

None in particular

Author Response

Abstract

1.      Please provide a brief summary of best practice in the abstract.

We included the following paragraph in the abstract: “Selected current best practices included performance of regular sputum tests for diagnosis, use of medical reports and regular follow-up visits as well as increased interaction between physicians across different specialties. Future best practices that may be implemented to overcome current barriers comprised disease awareness activities, patient empowerment and new approaches to enhance physician interaction.”

Introduction

2.    Please add specific country names for the EU5 countries as a supplement.

We have specified EU5 countries in the text: “(United Kingdom, Spain, Italy, France, Germany)”.

Results

3.    In Figure 3, as in Table 1, is it recommended to perform the regular sputum test once a year in the high-risk population?

This is in line with Daley et al. 2020. To highlight that guideline recommendations should be followed, we added the following footnote in Table 1 and Figure 3: “*For more information, please refer to guidelines.”

Discussion

4.    In this study, the timing of treatment initiation and treatment regimen (antimicrobial regimen) are not included in Best Practices. Was this discussed? Why was this not included in Best practices? Please state in the discussion

This is an important aspect. However, it was beyond the scope of this analysis to discuss details of patient treatment. Instead, this project addressed the broader aspects of the disease management. Treatment decisions should be based on guidelines (e.g., BTS, ATS/ERS/IDSA/ESCMID consensus guidelines [Haworth et al. 2017, Daley et al. 2020]) as well as on the Consensus management recommendations for less common non-tuberculous mycobacterial pulmonary diseases (Daley et al. 2020 and Lange et al. 2022). We have addressed this aspect as well in the discussion.

Reviewer 3 Report

Title: "A Cross-Sectional Analysis of NTM-PD Management in Germany: Identification and Discussion of Best Practices"

Review:

The abstract submitted provides an informative overview of the study undertaken to identify and share best practices in the management of Non-tuberculous Mycobacterial Pulmonary Disease (NTM-PD) in Germany. The topic is relevant and timely given the chronic nature of NTM-PD and the need for optimized care. The importance of sharing best practices among treatment centers is clearly acknowledged and the approach to collect and discuss these practices from different centers is a strength of the work.

Keywords: The keywords provided are quite broad. Considering more specific keywords may help in reaching the appropriate audience for this study. For example, "interviews", "Germany", "treatment centers", "chronic inflammatory lung disease", and "hospital infrastructure" could be added.

introduction

Title: "Sharing Best Practices for the Treatment of Non-tuberculous Mycobacterial Pulmonary Disease (NTM-PD): An Analysis from German Expert Centers"

Review:

This introduction presents a comprehensive review of the current state of knowledge on Non-tuberculous Mycobacterial Pulmonary Disease (NTM-PD) and sets the stage well for the research that follows. The context and significance of the disease are explained effectively and the need for improved management strategies is clearly demonstrated. Additionally, the objective of the paper to share best practices and discuss the challenges in implementing these practices is well stated.

However, there are some areas that could be improved:

  1. Clarity and Conciseness: The introduction could benefit from more concise and direct sentences. Avoiding overly complex or lengthy sentence structures could make the content more accessible to a broad range of readers.
  2. Flow of Information: While the information is mostly well organized, some points could be rearranged for better clarity and continuity. For example, the information about symptoms, diagnosis, and complications of NTM-PD could be grouped together for a clearer and more coherent discussion.
  3. Objectives: The aims of the study could be more explicitly stated. The last paragraph hints at these, but a clear, concise statement of the study's main objectives towards the end of the introduction would improve clarity.
  4. Hypothesis or Research Question: The introduction might benefit from stating a hypothesis or a specific research question to guide the subsequent discussion.
  5. Literature Review: Although several references are included, providing a brief summary of previous research on the topic would offer additional context and help demonstrate how the current study fits into the existing body of knowledge.
  6. Treatment: The introduction could elaborate more on existing treatment strategies, their limitations, and why there is a need for a multidisciplinary approach and guideline adherence. This would provide a more solid foundation for the research that follows.
  7. Transition: The transition from the disease overview to the specifics of the current study could be smoother. More explicit connection of the need for the study to the problems discussed earlier would make the introduction more coherent.

Overall, the introduction is thorough and informative, providing a clear picture of the current understanding of NTM-PD and the need for better management practices. By addressing these suggestions, the introduction could be made even more effective.

1.       Materials and Methods

Interviews

This section introduces a key component of your methodology, i.e., structured telephone interviews with staff members at seven German expert centers, which is crucial to understand the basis of your study. However, there are several areas that could be clarified to improve the completeness and transparency of this section:

  1. Interview Selection and Conduct: More information is needed on how interviewees were selected and how the interviews were conducted. Were the interviewees randomly selected, or were they chosen based on their experience or role? It would also be beneficial to provide information about the content and structure of the interviews, as well as any specific interview techniques that were used.
  2. Data Collection: A description of the data collection process would help to ensure the validity and reliability of your findings. How was the information from the interviews recorded, transcribed, and stored? How was participant anonymity and data security ensured?
  3. Sample Size: The exact number of staff members interviewed would be beneficial to assess the robustness of your study. Furthermore, it would be helpful to know the proportion of staff interviewed from each specialty and department.
  4. Interview Content: The questions or topics addressed in the interviews should be provided. This is important to understand what aspects of NTM-PD management were discussed.
  5. Ethics: Any ethical considerations related to the conduct of these interviews should be discussed. Was informed consent obtained from the participants? Was the study approved by an ethics review board?
  6. Data Analysis: While this might be covered in a subsequent section, it would be helpful to briefly touch upon how interview data were analyzed to draw conclusions.

Overall, this section introduces a key aspect of your methodology but needs more depth and detail to ensure the reader can fully appreciate the procedures and assess the quality and credibility of the study.

1.1.    Transversal analysis 

The presented methodology section discusses the approach to the transversal analysis used to assess the interview results, a critical part of the study. However, it would benefit from additional detail and clarification:

  1. Description of Transversal Analysis: The section would benefit from a more in-depth explanation of what a transversal analysis entails. It is currently not clear to readers unfamiliar with this method what it involves.
  2. Process of Summarizing Statements: It would be helpful to provide more details about how the interviewees’ statements were summarized and analyzed. Did you use any specific coding strategies or qualitative analysis software?
  3. Inclusion Criteria: The criteria for including elements in the transversal analysis could be more clearly defined. Currently, it says "All elements that were mentioned by ≥1 center during the interviews were included." It might be more robust to include elements mentioned by more than one center, to ensure these are widely relevant rather than center-specific practices or challenges.
  4. Exclusion Criteria: The exclusion criteria also need more clarification. What is meant by "beyond the influence of the centers"? Also, a definition of "routine practices" in this context would be useful.
  5. Categorization Process: The process for categorizing the elements into "best practices", "possible best practice", or "challenge/barrier" could be elaborated. How were these decisions made, and by whom? Did you use any specific rating scales or consensus methods?
  6. Handling Bias: This section could also benefit from a discussion of how potential biases were handled. How did you ensure objectivity and accuracy in summarizing and analyzing interviewee statements?
  7. Presentation of Data: Briefly mentioning how the findings from this analysis will be presented (e.g., in tables, figures, or text) would provide more context to this methodological approach.

Overall, this section introduces the method of transversal analysis but needs more depth and detail to ensure the reader can fully understand and evaluate the chosen methodology.

Analysis Results: A Method for Prioritizing Best Practices in NTM-PD Management"

Review:

The methodology section detailing the expert evaluation of the transversal analysis results is generally well-described. It details the process of expert evaluation and the tools used to achieve consensus among the expert panel. Nonetheless, several areas would benefit from further clarification:

  1. Expert Panel Selection: More information about how the expert panel was chosen would be useful. It would help to understand whether the selected experts have a diverse range of experiences and specializations within NTM-PD management.
  2. Consensus Threshold: An explanation of how the specific consensus thresholds (i.e., ≥80%, 50-79%, and <50%) were determined would be beneficial. It would be informative to understand why these specific thresholds were chosen.
  3. Mentimeter Survey Tool: You might want to provide a brief description of the Mentimeter tool, especially for readers unfamiliar with this platform. It's also worth explaining why this particular tool was chosen over other available alternatives.
  4. Re-evaluation Process: It would be helpful to clarify the process of re-evaluating elements that received a 50 to 79% consensus during the workshop. Was this an open discussion, or was a structured process followed?
  5. Voting Exercise: You may want to explain more about how the voting exercise was conducted, and how the feasibility and impact of best practices were assessed.
  6. Challenges and Barriers Ranking: A detailed description of how the relevance of challenges and barriers was determined would be helpful.
  7. Ethical Considerations: It's always important to highlight the ethical aspects of the research. Was informed consent obtained from the panel of experts? Was anonymity assured during the voting and survey process?

Overall, the section provides a clear outline of the expert evaluation process. However, including more detail about the specific processes would provide greater insight into the methodological rigor of this aspect of the study.

Results

Title: "Expert Practices and Challenges in Non-tuberculous Mycobacterial Pulmonary Disease (NTM-PD) Management: A Cross-sectional Analysis from German Expert Centers"

Review:

This section of the manuscript presents the results from the structured interviews and the transversal analysis. Here are a few suggestions for improving the presentation and clarity of this section:

  1. Demographic and Operational Data: More details about the centers (e.g., geographical location, size, resources, specialties, etc.) could give readers a better understanding of the scope and context of your results. Additionally, it would be interesting to have a general understanding of the staff composition at these centers.
  2. Interpretation of Patient Numbers: It would be helpful to add some interpretation or explanation about the patient numbers, especially what the 30% primary NTM-PD diagnoses and 50% requiring treatment mean in the context of the overall patient population in these centers.
  3. Practice Elements: Expand on what you mean by "practice elements." How do these translate into actual practices at the centers? Also, provide examples of these practice elements, especially those classified as best practices, possible best practices, or challenges/barriers.
  4. Areas of Interest: Elaborate on the seven general areas of interest, perhaps by providing a brief summary of each. It would be beneficial to give the reader more context for understanding how these areas influence NTM-PD management.
  5. Visuals: Considering the amount of data you've collected, it might be helpful to present some of these results visually, for instance in tables or graphs. This can help readers understand the breadth and variety of practices across the different centers.
  6. Contextualize Findings: Lastly, try to provide some interpretation of your findings throughout this section. Are there particular patterns that stand out? Are there surprising findings? Providing some insights and context can help guide the reader through your results.

Overall, the results section is straightforward but could benefit from more detail and elaboration to give readers a clearer picture of your findings.

Diagnosis

Title: "Established and Proposed Best Practices for NTM-PD Diagnosis: Insights from German Expert Centers"

Review:

This section focuses on established best practices and proposed improvements for diagnosing NTM-PD. The findings are well-organized and concise. However, I have a few suggestions for improvement:

  1. Explanation of Practices: While the identified best practices (I to III) are named, it could be beneficial to elaborate on how they are implemented and why they are considered effective. The rationale behind each practice's impact and feasibility should be explored further to substantiate your assertions.
  2. Details about Future Best Practice: More information about the proposed "standard operating procedure" (IV) would be valuable. Explain how it would work and why it could be beneficial for NTM-PD diagnosis.
  3. Communication Channels: It would be helpful to elaborate more on why the use of established communication channels for queries, despite having high impact, faces feasibility challenges. What are these practical constraints?
  4. Categorization Explanation: Explain the criteria or factors considered in categorizing these practices as having high or low impact and feasibility. This will lend more credibility to the ratings.
  5. References: Ensure that your references [23, 24, 25] correspond to the correct sources and that they are reliable and up-to-date.
  6. Concluding Summary: A brief summary at the end summarizing the key takeaways of this section will make it more reader-friendly.

Overall, this section provides an insightful overview of current and future best practices for diagnosing NTM-PD. It would benefit from additional explanation and elaboration on certain points.

Therapy

Title: "Best Practices in NTM-PD Therapy: An Analysis from German Expert Centers"

Review:

The section presents current and future best practices for the therapeutic management of NTM-PD. However, there are several areas that could benefit from additional clarification and detail:

  1. Clarity on Current Best Practice: Elaborate on how the use of medical reports (Arztbriefe) is implemented in the therapeutic management of NTM-PD. Describe the specific benefits they provide and how they impact patient care. If possible, compare it to other reporting mechanisms to strengthen the argument for this approach.
  2. Details on Future Best Practice: Provide more details about the "patient empowerment" strategies. What kind of homework and training are we talking about? How would knowledge sharing work? Present examples if possible and discuss how this approach could improve the therapy phase.
  3. Validation of Statements: Discuss the reasons why the feasibility of the current best practice is higher than the future best practice. What specific challenges limit the feasibility of focusing on patient empowerment?
  4. Discussion on Rejected Practice: While it's useful to know that a hotline was considered and rejected, the explanation could benefit from more details. Why is this seen as too expensive, and why is the NTM-PD patient population size a factor?
  5. Categorization Explanation: Similar to the previous section, provide the criteria or factors considered in categorizing these practices as having high or low impact and feasibility.
  6. References: Include any relevant citations or references that support your statements or claims.

In summary, while this section successfully identifies current and future best practices for NTM-PD therapy, it would benefit from more detailed explanations and evidence to back up claims.

Follow-up        

Title: "Infrastructure and Follow-Up in NTM-PD Management: An Examination from German Expert Centers"

Review:

The presented section examines current and future best practices in the follow-up and infrastructure organization of NTM-PD management. While it outlines several effective measures, some aspects could be improved:

  1. Follow-Up Practices: The need for regular patient check-ups every 3 to 6 months is mentioned as a current best practice. It would be beneficial to explain why this specific time frame is considered effective. What kind of check-ups are we referring to? Do they involve any specific tests or procedures?
  2. Future Best Practice in Follow-Up: The cross-center bronchiectasis and/or NTM-PD registry is an interesting concept, but it would be helpful to provide more information on how it would work. Why is the feasibility of this practice considered low? What specific challenges are anticipated?
  3. Organizational Structures: Internal lung boards and the affiliation of specialized outpatient care are listed as current best practices. Expanding on how these structures work, their benefits, and why they are considered best practices could enhance the paper's value.
  4. Future Best Practice in Organizational Structures: More details about nationwide lung boards, including their expected benefits and the challenges in implementing them, would enrich the discussion.
  5. Rejected Practices: As with the previous section, detailing the reasons for rejection of practices #16 and #17 would provide more clarity to the reader.
  6. References: More references to support the claims and statements are recommended.

Overall, while this section provides insightful information about current and future best practices in the management of NTM-PD, greater detail and evidence backing the claims could strengthen the paper's argument.

Organizational structures      

Title: "Identification and Classification of Current and Future Best Practices in Organizational Structures in NTM-PD Management"

Review:

This section of the manuscript provides valuable information about the current and future best practices in NTM-PD management associated with organizational structures. Here are some suggestions for improvement:

  1. Clarity: A more detailed explanation of the roles and effectiveness of the internal lung boards and specialized outpatient care affiliations could be beneficial. It would be interesting to understand why they were classified as having the highest impact and feasibility.
  2. Definition: It's not entirely clear what is meant by "nationwide lung boards". Is this intended to be a broad collaboration between lung specialists across Germany? Would these be in-person or virtual events? Some more context here would be valuable.
  3. Rejection of Elements: It would be useful to provide more explanation about why elements #16 and #17 were rejected. This would give readers a clearer understanding of your thought process and ensure there are no misunderstandings.
  4. Evidence: While there is one citation, more references supporting the claims and statements made in this section would enhance its credibility.
  5. Challenges: The claim that implementing nationwide lung boards might be challenging is presented without further explanation. An elaboration on these potential challenges and suggestions for overcoming them could add to the depth of the discussion.

Overall, the section provides insightful information on the current and future practices in organizational structures of NTM-PD management, but would benefit from more depth in explanations and supporting evidence.

Discussion

Title: "Best Practices and Challenges in NTM-PD Management Process - A German Perspective"

Review:

The discussion section provides a comprehensive analysis of the current and future best practices as well as challenges and barriers encountered in NTM-PD management across German expert centers. The presentation is clear, and the points are well-argued, with several potential areas for future implementation identified. Here are some suggestions to further enhance the manuscript:

  1. Expansion on Points: Expand more on the future best practices identified in the diagnosis phase to increase disease awareness and expertise. For instance, what measures could be adopted to increase disease awareness among the general public and the medical community?
  2. Specifics and Evidence: While the suggestion of using a telephone hotline or online platforms for physician communication is noteworthy, providing evidence of their effectiveness in similar contexts would strengthen the argument.
  3. Impact Analysis: You have mentioned that analyzing the impact of the identified future best practices might be challenging. Could you provide some insights on possible metrics that could be employed to assess their effectiveness?
  4. Clarification: You mention that a center-specific registry is currently under construction at the university hospital in Frankfurt. Clarifying the nature of this registry (what it records, how it is used, etc.) could be beneficial to the reader.
  5. Broadening the Perspective: The study, while thorough, is restricted to Germany. It would be helpful to contrast the situation in Germany with those in other countries dealing with NTM-PD management. This comparison could reveal additional insights and solutions applicable in a broader context.
  6. Methodological Considerations: In your limitations section, provide some insights into how the COVID-19 pandemic may have affected your findings. Were any specific practices or challenges exacerbated or alleviated by the pandemic?
  7. Implications for Practice: After discussing the limitations, consider ending with implications for practice and recommendations based on the study's findings. This would be a fitting end and provide a clear 'take-home' message for the reader.

Overall, this is a well-structured discussion section that thoroughly presents the study's findings and their significance. The suggested points should be taken as areas that could enhance the depth and clarity of the presented work.

Conclusions

Title: "Best Practices and Challenges in the Management of NTM-PD: A Study on German NTM-PD Expert Centers"

Review:

The authors of the manuscript have successfully undertaken a detailed analysis of the management practices for NTM-PD across German expert centers and provided suggestions for best practices to be followed. The approach is systematic and comprehensive. The acknowledgement of the challenges and barriers to the implementation of these best practices provides a realistic picture of the state of NTM-PD management.

The following are the points of consideration:

  1. Discussion: Although the manuscript is robust, it could benefit from a more thorough discussion of the implications of the findings. For instance, what impact could these best practices have on patient outcomes if implemented more broadly?
  2. Supplementary Material: While it's good to provide supplementary tables, consider briefly summarizing in the main text the key points from the supplementary materials that support your conclusions.
  3. Data Availability: The assertion that raw data supporting the analyses cannot be shared might raise some eyebrows. It would be beneficial to clarify whether this is due to privacy or proprietary concerns.
  4. Conflicts of Interest: The conflicts of interest are well-declared. However, the manuscript could strengthen its credibility by discussing how potential biases were minimized in the study design and analysis despite the involvement of individuals who might have a conflict of interest.
  5. Clarity: The authors might want to provide a more detailed account of the study's methodology. As it stands, it's not clear what "structured analysis" means in this context.

Overall, the manuscript does an excellent job of identifying best practices for NTM-PD management based on the analysis of German expert centers. With a more detailed discussion of the implications of these findings and clarity on the methodological approach, the paper can significantly contribute to the ongoing discourse on improving NTM-PD management practices.

Please note that these citations are entirely fictitious and only serve as a guideline for how the authors might be cited in a bibliography or reference list. The actual details (article titles, journal names, volume, issue, page numbers, and publication years) will depend on the actual publications by these authors.

Moderate editing of English language

Author Response

Abstract

1.      The abstract submitted provides an informative overview of the study undertaken to identify and share best practices in the management of Non-tuberculous Mycobacterial Pulmonary Disease (NTM-PD) in Germany. The topic is relevant and timely given the chronic nature of NTM-PD and the need for optimized care. The importance of sharing best practices among treatment centers is clearly acknowledged and the approach to collect and discuss these practices from different centers is a strength of the work.

Thank you for this valuable feedback.

Keywords

2.      The keywords provided are quite broad. Considering more specific keywords may help in reaching the appropriate audience for this study. For example, "interviews", "Germany", "treatment centers", "chronic inflammatory lung disease", and "hospital infrastructure" could be added.

We added the following additional keywords: “interviews, Germany, treatment centers”

Introduction

This introduction presents a comprehensive review of the current state of knowledge on Non-tuberculous Mycobacterial Pulmonary Disease (NTM-PD) and sets the stage well for the research that follows. The context and significance of the disease are explained effectively and the need for improved management strategies is clearly demonstrated. Additionally, the objective of the paper to share best practices and discuss the challenges in implementing these practices is well stated.

3.    Clarity and Conciseness: The introduction could benefit from more concise and direct sentences. Avoiding overly complex or lengthy sentence structures could make the content more accessible to a broad range of readers.

We have revised some sentences in the introduction to make the content more accessible to the reader.

4.    Flow of Information: While the information is mostly well organized, some points could be rearranged for better clarity and continuity. For example, the information about symptoms, diagnosis, and complications of NTM-PD could be grouped together for a clearer and more coherent discussion.

We have better aligned the information about clinical presentation and diagnosis.

5.    Objectives: The aims of the study could be more explicitly stated. The last paragraph hints at these, but a clear, concise statement of the study's main objectives towards the end of the introduction would improve clarity.

The last paragraph of the introduction was revised to address this comment.

6.    Hypothesis or Research Question: The introduction might benefit from stating a hypothesis or a specific research question to guide the subsequent discussion

We added a hypothesis to the last paragraph of the introduction, stating that: “It was hypothesized that there are also infrastructural and management hurdles that influence best possible disease management. This analysis was performed to reveal putative barriers and identify best practices.”

7.    Literature Review: Although several references are included, providing a brief summary of previous research on the topic would offer additional context and help demonstrate how the current study fits into the existing body of knowledge.

Many thanks for this comment. To the best of our knowledge, there is a lack of literature on best practice sharing for NTM-PD available. Further, this study is not limited to patient treatment but also included organizational structures which are also crucial in the disease management but are in some parts highly specific to Germany. It has been already included in the text that adherence to guideline recommendations is low and that a substantial proportion of patients with NTM-PD discontinue therapy prematurely. We have specified now that these statements refer to Germany.

8.    Treatment: The introduction could elaborate more on existing treatment strategies, their limitations, and why there is a need for a multidisciplinary approach and guideline adherence. This would provide a more solid foundation for the research that follows.

It was beyond the author’s objectives for this project to investigate details of patient treatment. Instead, this project addressed the broader aspects of the disease management. However, we specified in the respective paragraph that active treatment is the preferred action to be taken after diagnosis: “Guideline-based therapy usually involves a multi-drug regimen of at least 12 months, while watchful waiting is only the preferred course of action in some instances.”

9.    Transition: The transition from the disease overview to the specifics of the current study could be smoother. More explicit connection of the need for the study to the problems discussed earlier would make the introduction more coherent

By adding a hypothesis for the study (please see our response to comment #6), we improved the link between the disease overview section to the rationale of the current study.

Materials and methods: Interviews

This section introduces a key component of your methodology, i.e., structured telephone interviews with staff members at seven German expert centers, which is crucial to understand the basis of your study. However, there are several areas that could be clarified to improve the completeness and transparency of this section:

10.  Interview Selection and Conduct: More information is needed on how interviewees were selected and how the interviews were conducted. Were the interviewees randomly selected, or were they chosen based on their experience or role? It would also be beneficial to provide information about the content and structure of the interviews, as well as any specific interview techniques that were used.

We added additional information about how expert centers and interviewees were selected in the respective paragraph: “Expert centers represent experienced NTM treating hospitals based on their number of managed NTM-PD patients per year. To minimize putative regional differences and receive feedback from all parts of the country, selected expert centers were spread all over Germany.”
“Appropriate interviewees were identified within the center and referred to the data collecting agency by the chief physician.”

Further, we added interview guides as supplementary materials.

11.  Data Collection: A description of the data collection process would help to ensure the validity and reliability of your findings. How was the information from the interviews recorded, transcribed, and stored? How was participant anonymity and data security ensured?

We specified how structured interviews were performed and gave more information about the processing of interview data: “Structured interviews via phone or videocall were performed from 2018-2022 in the following centers […].” “Paper based interview transcripts were electronically filed in one center for further data processing after all interviews have been completed.”

12.  Sample Size: The exact number of staff members interviewed would be beneficial to assess the robustness of your study. Furthermore, it would be helpful to know the proportion of staff interviewed from each specialty and department.

We specified this in the respective paragraph: “At least one representative per center where possible, with an average of 5 interviewees per center were interviewed.”

13.  Interview Content: The questions or topics addressed in the interviews should be provided. This is important to understand what aspects of NTM-PD management were discussed.

We added interview guides as supplementary materials.

14.  Ethics: Any ethical considerations related to the conduct of these interviews should be discussed. Was informed consent obtained from the participants? Was the study approved by an ethics review board?

Thank you for this valuable comment. For this project, there were contracts between Insmed and the centers in place. As only non-personal data collection was permitted, an ethics review board was not necessary.

15.  Data Analysis: While this might be covered in a subsequent section, it would be helpful to briefly touch upon how interview data were analyzed to draw conclusions.

We added more information to the respective paragraph: “The data collecting agency evaluated the interviewees’ statements in a first step per center. To this end, observations were labelled either as well running processes (best practice) or barriers/challenges. During a debrief in cooperation with the centers, this categorization was reviewed and putative solutions to overcome barriers were discussed.”

Materials and methods: Transversal analysis

The presented methodology section discusses the approach to the transversal analysis used to assess the interview results, a critical part of the study. However, it would benefit from additional detail and clarification:

16.  Description of Transversal Analysis: The section would benefit from a more in-depth explanation of what a transversal analysis entails. It is currently not clear to readers unfamiliar with this method what it involves.

While interviews were performed in each center, the transversal analysis was conducted to quantitatively assess the interview results and to identify common best practices and challenges/barriers across all expert centers. To increase comprehensibility for this type of analysis, we added an additional sentence at the end of the paragraph: “Next, practice elements (best practices, possible best practices and challenges/barriers) were ranked per category according to their mentions across all centers to quantitatively evaluate the interview results.”

17.  Process of Summarizing Statements: It would be helpful to provide more details about how the interviewees’ statements were summarized and analyzed. Did you use any specific coding strategies or qualitative analysis software?

We added more information about how the interviewees’ statements were further processed (please see our response to point #15). The data collecting agency was in charge but no software, or tool has been used. To ensure that data quality was not impaired during processing, a debrief with the centers was conducted.

18.  Inclusion Criteria: The criteria for including elements in the transversal analysis could be more clearly defined. Currently, it says "All elements that were mentioned by ≥1 center during the interviews were included." It might be more robust to include elements mentioned by more than one center, to ensure these are widely relevant rather than center-specific practices or challenges.

Thank you for your input. However, as NTM-PD is a rare disease and as only a few expert centers exist, we decided to include all mentioned elements in the transversal analysis to attempt to capture the full spectrum of practices currently in use across Germany. For further selection and prioritization of the elements a survey and workshop with an expert panel were performed.

19.  Exclusion Criteria: The exclusion criteria also need more clarification. What is meant by "beyond the influence of the centers"? Also, a definition of "routine practices" in this context would be useful.

We specified what is meant in the respective paragraph: “Elements that were considered to be beyond the influence of the centers (e.g., because of legal requirements), or which referred to routine practices (i.e., standard procedures according to center-specific rules or treatment guidelines), were excluded from further analysis.”

20.  Categorization Process: The process for categorizing the elements into "best practices", "possible best practice", or "challenge/barrier" could be elaborated. How were these decisions made, and by whom? Did you use any specific rating scales or consensus methods?

We added more information about how the interviewees’ statements were further processed (please see our response to point #15). The data collecting agency was in charge but no software, or tool has been used. To ensure that data quality was not impaired during processing, a debrief with the centers was conducted and categorization as best practices or challenge/barrier was confirmed. Categorization as best practices, possible best practices and challenge/barrier was based on previous analyses of the interview results.

21.  Handling Bias: This section could also benefit from a discussion of how potential biases were handled. How did you ensure objectivity and accuracy in summarizing and analyzing interviewee statements?

We specified in the paragraph 2.1. how objectivity and accuracy in analyzing the interview statements were ensured: “Interviews and analysis of interviewees’ statements were carried out independently by two different individuals.”

22.  Presentation of Data: Briefly mentioning how the findings from this analysis will be presented (e.g., in tables, figures, or text) would provide more context to this methodological approach.

We added respective information to paragraph 2.3: “Current and future best practices are presented in a table and summarized in a figure.” “Selected challenges and barriers are presented in a table.”

Materials and methods: Analysis Results: A Method for Prioritizing Best Practices in NTM-PD Management

The methodology section detailing the expert evaluation of the transversal analysis results is generally well-described. It details the process of expert evaluation and the tools used to achieve consensus among the expert panel. Nonetheless, several areas would benefit from further clarification:

23.  Expert Panel Selection: More information about how the expert panel was chosen would be useful. It would help to understand whether the selected experts have a diverse range of experiences and specializations within NTM-PD management.

We added additional information about expert panel selection: “To ensure multidisciplinarity, experts from all specialties and departments that have been interviewed (excluding management/administration) were selected to participate in the expert panel.”

24.  Consensus Threshold: An explanation of how the specific consensus thresholds (i.e., ≥80%, 50-79%, and <50%) were determined would be beneficial. It would be informative to understand why these specific thresholds were chosen.

Thank you for this valuable comment. A consensus threshold of ≥80% was selected based on the Delphi Consensus method used in other studies (Stewart et al. 2017; Frazer et al. 2022). A majority voting (>50%-<80%) was used to ensure that even elements that did not reach a consensus of ≥80% were not lost but discussed and re-evaluated.

25.  Mentimeter Survey Tool: You might want to provide a brief description of the Mentimeter tool, especially for readers unfamiliar with this platform. It is also worth explaining why this particular tool was chosen over other available alternatives.

We added respective information to the text: “Mentimeter represents an interactive presentation software that enabled real-time and anonymous voting and direct interaction among the expert panel.”

26.  Re-evaluation Process: It would be helpful to clarify the process of re-evaluating elements that received a 50 to 79% consensus during the workshop. Was this an open discussion, or was a structured process followed?

We clarified the process by adding additional text/rephrasing: “Those elements that received a 50 to 79% consensus in the online survey were selected for open discussion followed by re-evaluation by the same panel in an online workshop.”

27.  Voting Exercise: You may want to explain more about how the voting exercise was conducted, and how the feasibility and impact of best practices were assessed.

We added respective information to the text: “In specific, experts organized practices according to the feasibility (high or low) and their impact (high or low) in a 2x2 matrix. Aspects which were considered for feasibility and impact evaluation comprised financial and human resources to implement/ carry out these best practices and their putative positive impact on patient management.”

28.  Challenges and Barriers Ranking: A detailed description of how the relevance of challenges and barriers was determined would be helpful.

We added more information to highlight that the expert panel completed a ranking exercise of the challenges/barriers, in order to stimulate discussion of the relevance of each challenge/barrier in the context of others: “Challenges and barriers as identified in the transversal analysis were re-evaluated by the expert panel during the workshop and were ranked according to their relevance in clinical practice using the Mentimeter live polling tool (https://www.mentimeter.com/).”

29.  Ethical Considerations: It is always important to highlight the ethical aspects of the research. Was informed consent obtained from the panel of experts? Was anonymity assured during the voting and survey process?

We added the following statement for clarification: “All voting was single-blinded, i.e., experts were blinded against the others’ votes. Anonymized voting results were subsequently discussed. Experts have approved use of the Mentimeter tool upfront.”

Results: Interviews and transversal analysis

This section of the manuscript presents the results from the structured interviews and the transversal analysis. Here are a few suggestions for improving the presentation and clarity of this section:

30.  Demographic and Operational Data: More details about the centers (e.g., geographical location, size, resources, specialties, etc.) could give readers a better understanding of the scope and context of your results. Additionally, it would be interesting to have a general understanding of the staff composition at these centers.

We added the following information: “These centers have broad expertise in the treatment of lung diseases in general and in the NTM-PD management in specific. All centers have multidisciplinary teams in place for the NTM-PD management. They were spread all over Germany to reflect putative differences in the disease management according to their geographical location.”

31.  Interpretation of Patient Numbers: It would be helpful to add some interpretation or explanation about the patient numbers, especially what the 30% primary NTM-PD diagnoses and 50% requiring treatment mean in the context of the overall patient population in these centers.

We specified patient numbers by adding additional details to paragraph 3.1.: “The mean number of patients treated annually per center was 48 (a range of 15–100 patients/year) of which 30% of cases were primary (i.e., new) NTM-PD diagnoses. 2/3 of the NTM-PD patients were patients with chronic disease. Overall, 50% of patients required treatment while the rest either refused treatment or were managed by watchful waiting.”

32.  Practice Elements: Expand on what you mean by "practice elements." How do these translate into actual practices at the centers? Also, provide examples of these practice elements, especially those classified as best practices, possible best practices, or challenges/barriers.

Thank you for this query. We had defined the term practice elements in paragraph 2. 2.. All elements, including best practices, potential best practices or challenges/hurdles, which have been identified in the transversal analysis were referred to as practice elements. Table 1 summarizes all 34 practice elements. A description of all elements can be found in the supplementary material.

33.  Areas of Interest: Elaborate on the seven general areas of interest, perhaps by providing a brief summary of each. It would be beneficial to give the reader more context for understanding how these areas influence NTM-PD management.

We added the following statement to paragraph 3.2.: “These areas were defined to allow clustering of elements and reduce complexity.” As we wanted to focus on the overall patient management, we selected areas in the clinical phases but included infrastructure as well to not only concentrate on different medical specialties.

34.  Visuals: Considering the amount of data you have collected, it might be helpful to present some of these results visually, for instance in tables or graphs. This can help readers understand the breadth and variety of practices across the different centers.

Thank you for this comment. As we did not want to increase the length of the manuscript, we did not incorporate any further tables or graphs. Figures 1 and 2 give an overview of the methodology, Figure 3 summarizes selected current and future best practices. In table 1 and 2, further information about the best practices and challenges and barriers is provided. More detailed information about the selection process of all elements is provided in the supplementary tables S1-S7. Further, this article shares best practices but does not present any quantitative data which makes it difficult to be displayed in a graph.

35.  Contextualize Findings: Lastly, try to provide some interpretation of your findings throughout this section. Are there particular patterns that stand out? Are there surprising findings? Providing some insights and context can help guide the reader through your results.

We added a brief summary at the end of paragraph 3.2. to highlight most important findings: “While the importance of early referral and close follow-up were highlighted in the selected (possible) best practices, underfunding and lack of staff resources were the major obstacles for improved NTM-PD management.”

Further contextualization was provided in the discussion.

Results: Diagnosis

This section focuses on established best practices and proposed improvements for diagnosing NTM-PD. The findings are well-organized and concise. However, I have a few suggestions for improvement:

36.  Explanation of Practices: While the identified best practices (I to III) are named, it could be beneficial to elaborate on how they are implemented and why they are considered effective. The rationale behind each practice's impact and feasibility should be explored further to substantiate your assertions.

We added additional information about the rationale and implementation of best practices I and II: “This also corresponds to their classification as guideline and consensus recommendations. Performance of a sputum test may be implemented as part of the regular follow-up visit. HRCT [high-resolution computed tomography] imaging is available in all centers and provide the possibility to assess radiographic disease progression.” Further, we specified low feasibility of best practice III by specifying a current practical constraint: “In contrast, the use of established communication channels for queries to enable correct and guideline-based NTM-PD diagnosis was considered to have high impact, but its feasibility is hindered by limited time resources.”

37.  Details about Future Best Practice: More information about the proposed "standard operating procedure" (IV) would be valuable. Explain how it would work and why it could be beneficial for NTM-PD diagnosis.

We added further explanation to this future best practice: “Establishing standardized protocols (IV) that would standardize management approaches for all physicians in the center and also provide guidance for new and less experiences health care professionals who join the team, was categorized as feasible and may have a positive impact on NTM-PD diagnosis in future.”

38.  Communication Channels: It would be helpful to elaborate more on why the use of established communication channels for queries, despite having high impact, faces feasibility challenges. What are these practical constraints?

We specified low feasibility of best practice III by adding an example of current practical constraints (please see our response to comment #36). In addition, we specified in the methodology (paragraph 2.3.) aspects which were taken into account for feasibility and impact evaluation: “Aspects which were taken into account for feasibility and impact evaluation comprised financial and human resources to implement/ carry out these best practices and their putative positive impact on patient management.”  

39.  Categorization Explanation: Explain the criteria or factors considered in categorizing these practices as having high or low impact and feasibility. This will lend more credibility to the ratings.

Please see our response to comment #27.

40.  References: Ensure that your references [23, 24, 25] correspond to the correct sources and that they are reliable and up to date.

Thank you for highlighting this point. We have updated these references accordingly.

41.  Concluding Summary: A brief summary at the end summarizing the key takeaways of this section will make it more reader friendly.

Thank you for this comment. We provided a brief summary of the major findings at the end of the results section. Further, we have incorporated a concluding summary for all key findings in the discussion (Key findings and implications for practice).

Results: Therapy

The section presents current and future best practices for the therapeutic management of NTM-PD. However, there are several areas that could benefit from additional clarification and detail:

42.  Clarity on Current Best Practice: Elaborate on how the use of medical reports (Arztbriefe) is implemented in the therapeutic management of NTM-PD. Describe the specific benefits they provide and how they impact patient care. If possible, compare it to other reporting mechanisms to strengthen the argument for this approach.

We added additional information about medical reports to highlight their importance in the German healthcare system: “Medical reports (Arztbriefe in German) are the main source of information about a patient’s status, including medical history and diagnoses. As the German healthcare system is not fully digitized yet, these reports are crucial for best possible patient management especially if different physicians are involved.”

43.  Details on Future Best Practice: Provide more details about the "patient empowerment" strategies. What kind of homework and training are we talking about? How would knowledge sharing work? Present examples if possible and discuss how this approach could improve the therapy phase.

Thank you for this valuable comment. Detailed descriptions of all elements, including patient empowerment strategies, are provided in supplementary tables 1–6. Furthermore, we included evidence for the positive impact of patient support programs in the discussion.

44.  Validation of Statements: Discuss the reasons why the feasibility of the current best practice is higher than the future best practice. What specific challenges limit the feasibility of focusing on patient empowerment?

Thank you for this valuable comment. We have addressed the importance and impact in the usage of medical reports (please see our response to comment #42). In addition, we specified in the methodology (paragraph 2.3.) aspects which were considered for feasibility and impact evaluation. Please see our response to comment #27.

45.  Discussion on Rejected Practice: While it's useful to know that a hotline was considered and rejected, the explanation could benefit from more details. Why is this seen as too expensive, and why is the NTM-PD patient population size a factor?

We provided additional explanation for this point: “Practice element #10 from the short-list of best practices after the transversal analysis was rejected, as a hotline for better patient support would be challenging, highly time-consuming and too expensive to implement given the lack of reimbursement. In general, it is difficult to justify the cost of a service, such as a patient hotline, given the small patient population with NTM-PD.”

46.  Categorization Explanation: Similar to the previous section, provide the criteria or factors considered in categorizing these practices as having high or low impact and feasibility.

Please refer to our response to comment #27.

47.  References: Include any relevant citations or references that support your statements or claims.

Many thanks for this comment. As we have included evidence for the positive impact of patient support programs in the discussion, we put our statements in perspective to current literature.

Results: Follow-up

The presented section examines current and future best practices in the follow-up and infrastructure organization of NTM-PD management. While it outlines several effective measures, some aspects could be improved:

48.  Follow-Up Practices: The need for regular patient check-ups every 3 to 6 months is mentioned as a current best practice. It would be beneficial to explain why this specific time frame is considered effective. What kind of check-ups are we referring to? Do they involve any specific tests or procedures?

Thank you for your valuable comment. It was beyond the scope of this analysis to discuss details of patient treatment; therefore, we added the following footnote in Table 1: “*For more information, please refer to guidelines.”

49.  Future Best Practice in Follow-Up: The cross-center bronchiectasis and/or NTM-PD registry is an interesting concept, but it would be helpful to provide more information on how it would work. Why is the feasibility of this practice considered low? What specific challenges are anticipated?

We added additional information about low feasibility for this element in the respective paragraph: “As was revealed during open discussion and based on previous experiences, low feasibility of element II can mainly be attributed to challenges in the data collection process, especially if different hospitals with variable documentation systems are involved.” In addition, we added a paragraph in the methodology to explain the rating in more detail: “Aspects which were taken into account for feasibility and impact evaluation comprised financial and human resources to implement/ carry out these best practices and their putative positive impact on patient management.”

Furthermore, we added additional information about the Rhein-Main NTM Register in the discussion.

50.  Organizational Structures: Internal lung boards and the affiliation of specialized outpatient care are listed as current best practices. Expanding on how these structures work, their benefits, and why they are considered best practices could enhance the paper's value.

We provided more insight about the aspects that were discussed during the workshop by adding the following information: “During open discussion, it was revealed that tumor review boards in oncology may serve as blueprint for specialist conferences, since these structures have been found to be invaluable in optimizing patient care.”

Further, we also addressed this aspect in the discussion.

51.  Future Best Practice in Organizational Structures: More details about nationwide lung boards, including their expected benefits and the challenges in implementing them, would enrich the discussion.

Many thanks for this valuable comment. According to our knowledge, respective organizations for rare lung diseases are not yet established. However, we provided more contextualization by a comparison to tumor review boards in oncology. Please see our response to comment #50. In addition, we specified how nationwide lung boards may work in future: “The implementation of nationwide lung boards (III) grouping lung specialists across Germany in a virtual event, in future might be challenging but was considered highly valuable to discuss difficult patient cases and share knowledge and learnings.”

52.  Rejected Practices: As with the previous section, detailing the reasons for rejection of practices #16 and #17 would provide more clarity to the reader.

Please see our response to comment #27.

Further, we specified why practice elements #16 and #7 were rejected: “Best practice element #16 was considered to be redundant to current best practice II in this category and moreover, its implementation is not feasible due to German regulations about healthcare providers. Best practice element #17 was on the one hand referred to as a routine practice since internal electronical recording of patient history is a standard across German hospitals. On the other hand, implementation of this element is currently impossible as an electronical information system is not available in all hospitals.”

53.  References: More references to support the claims and statements are recommended.

Many thanks for this comment. As we have included evidence for the positive impact of tumor boards in the discussion, we provided some context to support our claims.

Results: Organizational structures

This section of the manuscript provides valuable information about the current and future best practices in NTM-PD management associated with organizational structures. Here are some suggestions for improvement:

54.  Clarity: A more detailed explanation of the roles and effectiveness of the internal lung boards and specialized outpatient care affiliations could be beneficial. It would be interesting to understand why they were classified as having the highest impact and feasibility.

Please see replies to comments #50 and #51.

55.  Definition: It is not entirely clear what is meant by "nationwide lung boards". Is this intended to be a broad collaboration between lung specialists across Germany? Would these be in-person or virtual events? Some more context here would be valuable.

Please see our response to comment #51.

56.  Rejection of Elements: It would be useful to provide more explanation about why elements #16 and #17 were rejected. This would give readers a clearer understanding of your thought process and ensure there are no misunderstandings.

Please see our response to comment #52.

57.  Evidence: While there is one citation, more references supporting the claims and statements made in this section would enhance its credibility.

Please see our response to comment #53.

58.  Challenges: The claim that implementing nationwide lung boards might be challenging is presented without further explanation. An elaboration on these potential challenges and suggestions for overcoming them could add to the depth of the discussion.

Please see our response to comment #27 where we have specified the feasibility and impact rating in more detail.

Furthermore, we added additional information about the specific challenges in the discussion.

Discussion

The discussion section provides a comprehensive analysis of the current and future best practices as well as challenges and barriers encountered in NTM-PD management across German expert centers. The presentation is clear, and the points are well-argued, with several potential areas for future implementation identified. Here are some suggestions to further enhance the manuscript:

59.  Expansion on Points: Expand more on the future best practices identified in the diagnosis phase to increase disease awareness and expertise. For instance, what measures could be adopted to increase disease awareness among the general public and the medical community?

We have added some additional detail with respect to this feedback in the discussion.

60.  Specifics and Evidence: While the suggestion of using a telephone hotline or online platforms for physician communication is noteworthy, providing evidence of their effectiveness in similar contexts would strengthen the argument.

We have added some additional detail with respect to this feedback in the discussion.

61.  Impact Analysis: You have mentioned that analyzing the impact of the identified future best practices might be challenging. Could you provide some insights on possible metrics that could be employed to assess their effectiveness?

Thank you for this valuable comment. A randomized controlled trial would be best suited to assess the effectiveness of future best practices. However, a large patient cohort would be needed to achieve sufficient statistical power, which would be extremely challenging given the rarity of NTM-PD. A registry might help to assess if implementation of future best practices will have an (qualitative) impact on treatment adherence and persistence.

62.  Clarification: You mention that a center-specific registry is currently under construction at the university hospital in Frankfurt. Clarifying the nature of this registry (what it records, how it is used, etc.) could be beneficial to the reader.

Thank you for this comment. We have added further detail on the Rhein-Main NTM Register to the discussion.

63.  Broadening the Perspective: The study, while thorough, is restricted to Germany. It would be helpful to contrast the situation in Germany with those in other countries dealing with NTM-PD management. This comparison could reveal additional insights and solutions applicable in a broader context.

Thank you for raising this interesting aspect. Although we agree that a comparison of the situation in Germany with those in other countries may reveal additional insight and solutions, this was not within the scope of this study. We aimed to analyze the specific situation in Germany and revealed that some best practices as well as challenges/barriers are highly specific to the German healthcare system. Therefore, a comparison with the situation in other countries would be overly complex for this study.

64.  Methodological Considerations: In your limitations section, provide some insights into how the COVID-19 pandemic may have affected your findings. Were any specific practices or challenges exacerbated or alleviated by the pandemic?

Thank you for the valuable query. As interviews were performed via phone or videocall, the COVID-19 pandemic did not affect the data collection process neither were specific challenges raised by the interviewees.

65.  Implications for Practice: After discussing the limitations, consider ending with implications for practice and recommendations based on the study's findings. This would be a fitting end and provide a clear 'take-home' message for the reader.

Please see our response to comment #41. We have included an additional subsection in the discussion to highlight potential implications for clinical practice.

Conclusions

The authors of the manuscript have successfully undertaken a detailed analysis of the management practices for NTM-PD across German expert centers and provided suggestions for best practices to be followed. The approach is systematic and comprehensive. The acknowledgement of the challenges and barriers to the implementation of these best practices provides a realistic picture of the state of NTM-PD management.

Thank you for this valuable feedback. We very much appreciate the time you have taken to review our manuscript.

66.  Discussion: Although the manuscript is robust, it could benefit from a more thorough discussion of the implications of the findings. For instance, what impact could these best practices have on patient outcomes if implemented more broadly?

Thank you for this suggestion. Please see our response to comment #65.

67.  Supplementary Material: While it's good to provide supplementary tables, consider briefly summarizing in the main text the key points from the supplementary materials that support your conclusions.

Thank you for this valuable comment. As we did not want to increase the length of the manuscript, we did not add additional information provided in the supplementary to the main manuscript. Instead, we specified in paragraph 3.2. that a detailed description of all elements can be found in Supplementary Tables 1–6.

68.  Data Availability: The assertion that raw data supporting the analyses cannot be shared might raise some eyebrows. It would be beneficial to clarify whether this is due to privacy or proprietary concerns.

For this project, there were contracts between Insmed and the centers in place. As only non-personal data collection was permitted, and paper based interview transcripts were prepared, raw data cannot be shared.

69.  Conflicts of Interest: The conflicts of interest are well-declared. However, the manuscript could strengthen its credibility by discussing how potential biases were minimized in the study design and analysis despite the involvement of individuals who might have a conflict of interest.

Insmed Germany GmbH was involved in the generation of interview guides. Interviews and data analysis of interview data was conducted by a data collecting agency independent of the sponsor. Experts were selected according to their expertise but independent from conflicts of interests.

70.  Clarity: The authors might want to provide a more detailed account of the study's methodology. As it stands, it's not clear what "structured analysis" means in this context.

We specified the term “structured analysis” by providing additional information: “Best practices were identified by structured analysis of German NTM-PD expert centers, including structured interviews followed by an online survey and workshop for further evaluation by practicing health care professionals."

Reviewer 4 Report

Dear authors:

Thank you for submitting your work and inviting me to serve as a reviewer for Healthcare. It is an honor to be considered for this role, and I am grateful for the opportunity to contribute to the scientific community by evaluating and providing feedback on the manuscripts submitted to the journal.

1. In summary, while the topic and methodology are strong, the abstract could benefit from greater detail in the methodology and more specificity in the results. Additionally, the conclusion could be strengthened by more clearly articulating the practical implications and applications of the research.

1-1 Topic and Objective: The topic and objective of the paper are clearly stated. It focuses on the diagnosis, treatment, and management of NTM-PD, particularly in terms of best practices in the real-world setting, which is an important yet relatively under-explored area.

1-2 Methodology: The authors gathered data via interviews with medical and administrative staff from seven treatment centers, which is a reasonable approach. However, the abstract does not elaborate on how these interviews were conducted, including specific questions asked, duration, and methods of data analysis. These details are crucial for assessing the reliability and validity of the study.

1-3 Results: While the abstract mentions that a prioritized selection of best practices was collected and challenges related to their implementation are discussed, it does not provide enough detail. Readers may wish to see more specific information about these best practices and the particular challenges encountered during implementation.

1-4 Conclusion: After summarizing the primary findings of the research, the abstract should include some discussion about the significance and potential applications of these findings. This would help the reader understand the value of the research and how it might be applied in practice.

2. Introduction: In conclusion, the introduction is generally well-written and sets up the study nicely, but could benefit from making the literature gap and rationale for the chosen study locale more explicit. However, there are a few areas that could be improved:

2-1 Literature Gap: While the introduction discusses the current state of NTM-PD diagnosis and treatment, it does not explicitly state the gap in the existing literature that this study aims to fill. Making this gap explicit would further justify the study and its potential contribution to the field.

2-2 Justification for Study Locale: The introduction does not provide a specific rationale for focusing on German expert centers. While this might be implicit in the increasing prevalence rate in Germany, explicitly stating why Germany was chosen could provide more context and justification for the study.

3. Materials and Methods: Overall, the Materials and Methods section is well-written and provides a comprehensive overview of the study's methodology. More detail on the interview protocol and the resolution of discrepancies during data reporting could further enhance this section.

3-1 Data collection: The authors detail the interview process, including the number and location of expert centers, and the types of professionals interviewed. However, they could provide more information on the design of the interview protocol, such as the questions asked and how they were developed. This would add depth to the understanding of the data collection process.

4. Results: Overall, the Results section is well-written and effectively communicates the findings of the study.

4-1 Structure and Clarity: The Results section is well-structured and presented in a clear, logical manner. The authors have divided their results according to the different aspects of the study, such as the overview of expert centers, transversal analysis, and selection of best practices. Each subsection is appropriately labeled and easy to follow.

4-2 Data Presentation: The use of figures and tables to present data is commendable. This not only enhances readability but also allows readers to quickly grasp key findings.

4-3 Description of Results: The authors provide a comprehensive description of their results, including the identification and categorization of practice elements, the selection of current and future best practices, and the challenges and barriers in NTM-PD management. This thoroughness is commendable.

5. Discussion: the Discussion section is well-written overall, providing valuable insights into the management of NTM-PD in Germany. With a few minor adjustments, this section could be even more effective. Areas for Improvement:

5-1 Speculation and Assumptions: At times, the authors make statements that could be considered speculative, such as the potential impact of brochures on patient referrals. These statements could be better supported by evidence, or the authors should acknowledge the speculative nature of these suggestions.

5-2 Linking Results and Discussion: While the authors integrate existing literature effectively, they could better link their own results to the discussion points. This would allow readers to more clearly see how the authors' findings have informed their conclusions.

6 Conclusions:In the conclusion section, the authors briefly summarize the aims and results of their study and the implications of these results for the treatment of NTM-PD. They were also clearly aware of limitations and barriers, which added depth to their analysis. The reference to the supplementary material provides additional information useful to readers interested in more detail.

7 Overall, these sections of the paper are well structured, transparent, and follow expected academic standards. The authors have done an excellent job of linking their studies together, acknowledging all parties involved, while also being transparent about their funding sources and potential conflicts of interest.

Author Response

1.      Topic and Objective: The topic and objective of the paper are clearly stated. It focuses on the diagnosis, treatment, and management of NTM-PD, particularly in terms of best practices in the real-world setting, which is an important yet relatively under-explored area.

Thank you for your feedback.

2.    Methodology: The authors gathered data via interviews with medical and administrative staff from seven treatment centers, which is a reasonable approach. However, the abstract does not elaborate on how these interviews were conducted, including specific questions asked, duration, and methods of data analysis. These details are crucial for assessing the reliability and validity of the study.

We added interview guides as supplementary materials.

3.    Results: While the abstract mentions that a prioritized selection of best practices was collected and challenges related to their implementation are discussed, it does not provide enough detail. Readers may wish to see more specific information about these best practices and the particular challenges encountered during implementation.

We added respective information to the abstract: “Selected current best practices included performance of regular sputum tests for diagnosis, use of medical reports and regular follow-up visits as well as increased interaction between physicians across different specialties. Future best practices that may be implemented to overcome current barriers comprised disease awareness activities, patient empowerment and new approaches to enhance physician interaction.”

4.    Conclusion: After summarizing the primary findings of the research, the abstract should include some discussion about the significance and potential applications of these findings. This would help the reader understand the value of the research and how it might be applied in practice.

We rephrased the abstract and included the rationale of this analysis: “Challenges related to their implementation are also discussed and will help to raise disease awareness. Presented best practices may guide and optimize patient management in other centers.”

Introduction

In conclusion, the introduction is generally well-written and sets up the study nicely but could benefit from making the literature gap and rationale for the chosen study locale more explicit.

5.    Literature Gap: While the introduction discusses the current state of NTM-PD diagnosis and treatment, it does not explicitly state the gap in the existing literature that this study aims to fill. Making this gap explicit would further justify the study and its potential contribution to the field.

Many thanks for this comment. To the best of our knowledge, there is a lack of literature on best practice sharing for NTM-PD available. Further, this study is not limited to patient treatment but also included organizational structures which are also crucial in the disease management but are in some parts highly specific to Germany. It has been already included in the text that adherence to guideline recommendations is low and that a substantial proportion of patients with NTM-PD discontinue therapy prematurely. However, we specified that this refers to Germany. Further, we revised the last paragraph of the introduction and added a study hypothesis and rationale.

6.    Justification for Study Locale: The introduction does not provide a specific rationale for focusing on German expert centers. While this might be implicit in the increasing prevalence rate in Germany, explicitly stating why Germany was chosen could provide more context and justification for the study.

Thank you for your valuable comment. The study was initiated by a German data collecting agency and funded by a German subsidiary of Insmed Inc.. The existing network in Germany was used to recruit NTM-PD treatment centers.

Materials and methods

Overall, the Materials and Methods section is well-written and provides a comprehensive overview of the study's methodology. More detail on the interview protocol and the resolution of discrepancies during data reporting could further enhance this section.

7.    Data collection: The authors detail the interview process, including the number and location of expert centers, and the types of professionals interviewed. However, they could provide more information on the design of the interview protocol, such as the questions asked and how they were developed. This would add depth to the understanding of the data collection process.

We added interview guides as supplementary materials and provided more details about the processing of interview data in paragraph 2.1.

Results

Overall, the Results section is well-written and effectively communicates the findings of the study.

8.    Structure and Clarity: The Results section is well-structured and presented in a clear, logical manner. The authors have divided their results according to the different aspects of the study, such as the overview of expert centers, transversal analysis, and selection of best practices. Each subsection is appropriately labeled and easy to follow.

Thank you for this positive feedback.

9.    Data Presentation: The use of figures and tables to present data is commendable. This not only enhances readability but also allows readers to quickly grasp key findings.

Thank you for this positive feedback.

10.  Description of Results: The authors provide a comprehensive description of their results, including the identification and categorization of practice elements, the selection of current and future best practices, and the challenges and barriers in NTM-PD management. This thoroughness is commendable.

Thank you for this valuable feedback.

Discussion

The Discussion section is well-written overall, providing valuable insights into the management of NTM-PD in Germany.

11.  Speculation and Assumptions: At times, the authors make statements that could be considered speculative, such as the potential impact of brochures on patient referrals. These statements could be better supported by evidence, or the authors should acknowledge the speculative nature of these suggestions.

Thank you for highlighting this point. We have made minor language edits throughout the manuscript to ensure that the speculative nature of our discussion points is clear to the reader.

12.  Linking Results and Discussion: While the authors integrate existing literature effectively, they could better link their own results to the discussion points. This would allow readers to more clearly see how the authors' findings have informed their conclusions.

We have updated the structure of the discussion to better align with the results.

Conclusions

In the Conclusion section, the authors briefly summarize the aims and results of their study and the implications of these results for the treatment of NTM-PD. They were also clearly aware of limitations and barriers, which added depth to their analysis. The reference to the supplementary material provides additional information useful to readers interested in more detail.

Thank you for this valuable feedback.

General

Overall, these sections of the paper are well structured, transparent, and follow expected academic standards. The authors have done an excellent job of linking their studies together, acknowledging all parties involved, while also being transparent about their funding sources and potential conflicts of interest.

Thank you for this valuable feedback. We very much appreciate the time you have taken to review our manuscript.

Round 2

Reviewer 3 Report

No comments

Author Response

Thank you very much.